# Human Pose Estimation via Parse Graph of Body Structure

## Abstract

When observing a person's body, humans can extract the structured representation of the body called a parse graph, which includes the hierarchical decompositions from the entire body to parts and primitives and the context relations by horizontal links between the body parts. This ability helps humans better locate body structures at different levels. In order for the model to have this ability for human pose estimation (HPE), we design a hierarchical network to model the context relations and hierarchical structure in the parsing graph by convolutional neural networks. It overcomes the problem that most methods ignore context relations in the inference of hierarchical structure for HPE. Our network contains bottom-up and top-down stages. In the bottom-up stage, the structural features of the hierarchy are captured from primitives to parts and the entire body. Then in the top-down stage, with the context information of each body part, the structural features of the body parts are refined separately rather than together from the entire body to parts and primitives. Experiments show that our model enhances the reasonableness of predictions and achieves superior results on the COCO keypoint detection and MPII human pose datasets.

## 1 Introduction

The main task of 2D human pose estimation (HPE) is to locate the joints position of each person from a single image, such as wrists, ankles, knees, etc. It is divided into single-person pose estimation and multi-person pose estimation. We focus on single-person pose estimation.

When watching a person's body, humans can quickly extract its structured representation, which can be represented by a parse graph. It enables humans to better finish the task of HPE. The parse graph includes hierarchical structure in vertical direction and context

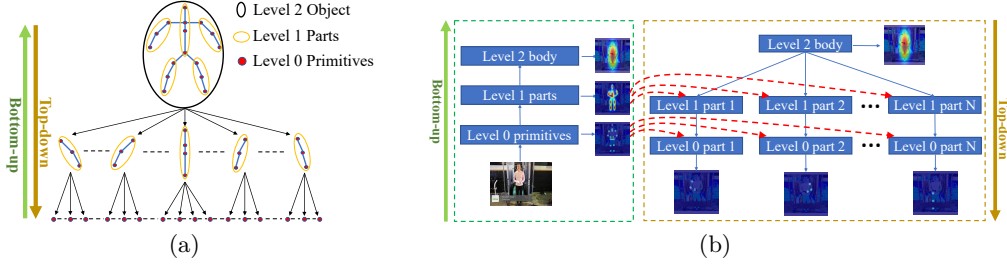

Figure 1: (a) The parse graph of body structure. It includes a tree-structured decomposition in vertical arrows and context relations in horizontal arrows. (b) Overview of our method. It includes bottom-up and top-down two stages. In the bottom-up stage, the network learns hierarchical structure features supervised by heatmaps. In the top-down stage, the context information and coarse prediction results of each body part is obtained from the bottom-up stage (horizontal red dotted line).

relations in horizontal direction (Zhu et al., 2007). The hierarchical structure is a tree structure that represents the hierarchical decompositions from the entire body to parts and primitives, which allows models to capture high-order relationships between parts (Tian et al., 2012) and prevents models over-fit due to the existence of structural imformation (Zhu et al., 2007). The context relations is the relation between parts at the same level such as spatial and functional relations, which ensures good spatial relationships between parts at the same level (Zhu et al., 2007).

The hierarchical structure and the context relations have been used for some tasks, such as object detection (Chen et al., 2014; Sudderth et al., 2005; Zhu et al., 2010), segmentation (Ding et al., 2023) and HPE (Tang et al., 2018; Yang & Ramanan, 2012; Chen & Yuille, 2014; Chu et al., 2016b; Tompson et al., 2014; Zhang et al., 2019; Yang et al., 2016; Chu et al., 2016a; De Bem et al., 2018; Tian et al., 2012; Wang et al., 2011; Rothrock et al., 2013; Sun & Savarese, 2011; Park et al., 2017). However, most of them (Chen et al., 2014; Zhu et al., 2010; Ding et al., 2023; Tang et al., 2018; Yang & Ramanan, 2012; Chen & Yuille, 2014; Chu et al., 2016b; Tompson et al., 2014; Zhang et al., 2019; Yang et al., 2016; Chu et al., 2016a; De Bem et al., 2018; Tian et al., 2012; Sun & Savarese, 2011) always ignore one of the Hierarchical structure and the context relations or learn them implicitly. Tang et al. (Tang et al., 2018) argue that some method (Wang et al., 2011; Rothrock et al., 2013; Sun & Savarese, 2011; Park et al., 2017; Johnson & Everingham, 2010) have difficulty describing the complex compositional relationships among body parts due to some unreasonable assumptions. Moreover, Some methods use simple algorithms to model the hierarchical structure and the context relations, which has poor performance in complex scenarios. For example, Chen et al. (Chen et al., 2014) uses segDPM (Fidler et al., 2013) to model various parts and Sudderth et al. (Sudderth et al., 2005) uses Gibbs sampler to learning the parameters of hierarchical probabilistic model. Fortunately, in recent years, many methods (Tang et al., 2018; De Bem et al., 2018; Zhuang et al., 2019; Ai et al., 2017; Jun et al., 2020) make appearance representations for body parts and use them for supervision of convolutional neural networks (CNNs). In this way, the hierarchical structure and the context relations can be modeled easily.

To solve the above issues, we design a hierarchical network for HPE based on the parse graph of body structure. The parse graph of body structure we designed is shown in Fig. 1(a), it contains a hierarchical structure and context relations. With the parse graph, the information about the entire body can be obtained by recursively predicting the state of their sub-parts in a bottom-up way, and then, with context relations, low-level parts can be refined by updating high-level parts states in advance in a top-down way. This global adjustment enables pose estimates to optimally satisfy the constraints of hierarchical and context relations.

An overview of our approach as shown in Fig. 1(b), which includes bottom-up/top-down inference stages. In the bottom-up stage, the features and coarse prediction results of hierarchical structures are obtained. More importantly, in the top-down stage, different from other methods, our network learns the features of the human body structure by grouping so multiple branches structure is adopted, which can avoid over-fit. Each branch obtains corresponding context information and coarse prediction results from the previous stage to help refine the corresponding part. Our main contributions are as follows:

- We design a novel and valid architecture to learn the hierarchical structure and context relations in the parse graph via CNNs.

- We propose a novel approach to model context relations for each body part and obtain the context information of each body part to help refine them.

- Our proposed approach is highly interpretable, and the effectiveness of our method is demonstrated on the COCO keypoint detection (Lin et al., 2014), MPII human pose (Andriluka et al., 2014) and Crowdpose (Li et al., 2019) datasets.

## 2 Related work

**Parse graphs**. As shown in Fig. 1(a), the parse graph includes a tree-structured decomposition in the vertical direction and context relations, such as spatial and functional relations, in the horizontal direction (Zhu et al., 2007). Some methods (Han & Zhu, 2008; Wu & Zhu, 2011; Zhu et al., 2007) parse images into their corresponding visual patterns and make inference through bottom-up and top-down stages, and are exploited in HPE (Yang & Ramanan, 2012; Chen & Yuille, 2014; Chu et al., 2016b; Tompson et al., 2014; Zhang et al., 2019; Yang et al., 2016; Chu et al., 2016a; Tian et al., 2012; Wang et al., 2011; Rothrock et al., 2013; Sun & Savarese, 2011; Park et al., 2017). But prior methods fail to perform well due to simple modeling for hierarchical structure. Although this problem has been recently addressed by the method (Tang et al., 2018; De Bem et al., 2018), it is easy to over-fit due to shared features for all structures at the same level and ignores the modeling of context relations. According to the parse graph, we learn unique features for each body part and design a multiscale context module to obtain context relations and information of each body part through the cosine similarity between structural feature maps.

**Human pose estimation**. With the rapid development of CNNs, many excellent HPE methods based on CNNs have been designed by researchers, and lots of great backbones have emerged, such as CPM (Wei et al., 2016), Hourglass Network (Newell et al., 2016), FPN (Yang et al., 2017a), CPN (Chen et al., 2018), Simple Baseline (Xiao et al., 2018), HRNet (Sun et al., 2019) and RSN (Cai et al., 2020). However, these models' interpretability becomes especially difficult as their backbones become more complex. In contrast, our model is simply extended based on HRNet and uses heatmaps at different levels as intermediate supervision at different layers of the network so the interpretability of our model is stronger.

**Appearance representations**. There are three kinds of appearance representations, namely joint, skeleton, and body. (i) Joint appearance. Many methods (Sun et al., 2019; Xiao et al., 2018; De Bem et al., 2018; Tang et al., 2018; Wang et al., 2019) generate isotropic Gaussians centered at each joint. (ii) Skeleton appearance. Jun et al. (Jun et al., 2020) generate a binary image, where the pixel corresponding to the line connecting joint $i$ and $j$ is 1, and the other pixels are 0. Ai et al. (Ai et al., 2017) draw a line $p$ connecting each pair of joints, and then use it as the center of a Gaussian distribution. De Bem et al. (De Bem et al., 2018) use the midpoint of each pair of joints as the center of the Gaussian distribution. (iii) Entire body appearance. De Bem et al. (De Bem et al., 2018) use the mean of annotated joint centers as body centers and also as Gaussian distribution centers. In our work, Gaussian distribution heatmaps are made for the network supervision.

## 3 Our approach

In this section, we will detail how to design the hierarchical network for HPE. First, we formulate the parse graph of body structure. Second, we will elaborate on the specific method of network construction. Thirdly, we describe how to generate appearance representations. Finally, we illustrate how to design multiscale context and prior fusion modules.

### 3.1 Parse graphs

The parse graph of body structure is shown in Fig. 1(a) and it consists of a 4-tuple $(V, E, \phi^{and}, \phi^{leaf})$. $(V, E)$ and $(\phi^{and}, \phi^{leaf})$ represent the parse graph structure and potential function respectively, where $V = V^{and} \cup V^{leaf}$. $V^{and}$ combine sub-parts into high-level parts according to certain rules (Zhu et al., 2007), $V^{leaf}$ represents the lowest-level parts and $E$ represents the edge of the parse graph. For HPE, A state variable $w_u$ can be represented by position $p_u$ and type $t_u$: $w_u = \{p_u, t_u\}, u \in V$. Let $\Theta$ denote the set of all state variables in the model, and it can be expressed as:

$$P(\Theta|I) = \frac{1}{K} exp\{-T(\Theta, I)\} \tag{1}$$

where $T(\Theta, I)$ is the energy function, $K$ is the partition function, and $I$ is the input image. For convenience, we discard $I$ and let $G(\Theta) = -T(\Theta, I)$. $G(\Theta)$ is expressed as:

$$G(\Theta) = \sum_{u \in V^{leaf}} \phi_u^{leaf}(w_u, I) + \sum_{u \in V^{and}} \phi_u^{and}(w_u, \{w_v\}_{v \in ch(u)}) \tag{2}$$

where $ch(u)$ denotes the set of children of node $u$. The work of first term is a detector. The second denotes the higher-order potential function measuring the state compatibility among part $u$ and its child parts $\{v : v \in ch(u)\}$.

Using the parse graph, the optimal state of the input image $I$ can be efficiently calculated in bottom-up and top-down ways. The maximum score $G(\Theta)$ in the bottom-up stage can be formulated as:

$$(Leaf)G_u^{\uparrow} = \phi_u^{leaf}(w_u, I) \tag{3}$$

$$(And)G_u^{\uparrow} = \sum_{v \in ch(u)} \max_{w_v}[\phi_{u,v}^{and}(w_u, w_v) + G_v^{\uparrow}(w_v)] \tag{4}$$

where $G_u^{\uparrow}$ is the maximum value of the subgraph composed of node $u$ and all its subgraphs $v$, the state of the node $u$ is $w_u$, and Eq. 3 is the boundary condition.

The optimal state for node v in the top-down stage, can be formulated as:

$$(Root)w_u^* = argmax_{w_u}G_u^{\downarrow}(w_u) \equiv argmax_{w_u}G_u^{\uparrow}(w_u) \tag{5}$$

$$(Non-root)w_v^* = argmax_{w_v}G_v^{\downarrow}(w_v) \equiv argmax_{w_v}[\phi_{u,v}^{and}(w_u^*, \{w_h\}_{h \in V_s}) + G_v^{\uparrow}(w_v)] \tag{6}$$

where node $u$ in Eq. 5 is the only parent node of node $v$, $G_u^{\uparrow}(w_u)$ and $G_v^{\uparrow}(w_v)$ are acquired from the bottom-up stage, $G_u^{\downarrow}(w_u)$ and $G_v^{\downarrow}(w_v)$ represent the refinement score graphs of nodes $u$ and $v$ respectively, and $w_u^*$ and $w_v^*$ are the optimal states of u and v respectively. Especially, $V_s$ is a set containing all nodes of the same level as node $v$, including $v$. In other words, context relations between node $v$ and other nodes at the same level need to be considered when refining the score graph of node $v$. Eq. 5 is the boundary condition.

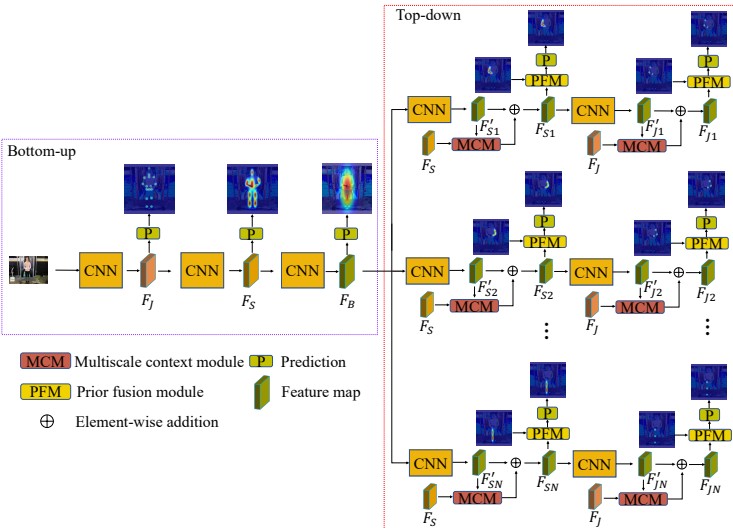

Figure 2: The framework of our method. In the top-down stage, the context information is obtained by filtering and combining the feature maps $F_S$ or $F_J$ through the multiscale context module. In addition, the prior fusion module is used to fuse the coarse prediction results from the bottom-up stage with corresponding refinement branch in the top-down stage.

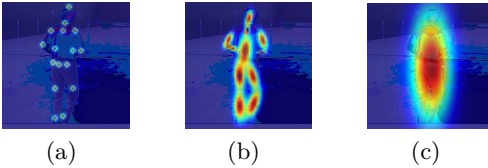

(a)      (b)      (c)

Figure 3: The 2D Gaussian heatmaps of each structure. For display convenience, we add together the heatmaps of all structures at the same level. (a), (b) and (c) are the 2D Gaussian heatmap of joints, skeleton and body respectively.

## 3.2 Network architecture

Our network follows the bottom-up and top-down learning method (Tang et al., 2018). The difference is that in the top-down stage, we use different branches to refine each structure body part and more importantly model context relations and obtain context information.

The framework of our method is shown in Fig. 2, where the yellow CNN module is the convolution and multiscale fusion operations in HRNet. In the bottom-up stage, we obtain different levels of structural features $F_J$, $F_S$ and $F_B$ through supervision and the context information of each joint and skeleton part is obtained from $F_J$ and $F_S$ respectively through the multiscale context module. In the top-down stage, there are $N$ branches, where $N = 5$, refining different parts of body respectively. Each branch contains two modules, namely multiscale context and prior fusion modules. The multiscale context module allows each branch to utilize features of other body parts to help the refinement. The prior fusion module combines the coarse prediction results from the bottom-up stage and corresponding branches of the top-down stage to help the refinement. The two modules will be introduced in detail later. We use the prediction results of each branch in the top-down stage as the final result for HPE.

## 3.3 Appearance representations

As shown in Fig. 3, we use Gaussian heatmaps to represent structural appearance for the supervision of our network following details of how we generate heatmaps of different hierarchies.

**Primitives**. We follow the method of HRNet to generate isotropic Gaussian distribution heatmaps for each joint. Let $J_i$ represent the i-th joint, the Gaussian heatmap of $J_i$ is:

$$H_{J_i} = e^{-\frac{(X - P_{ix})^2 + (Y - P_{iy})^2}{2\sigma_i^2}}$$ (7)

where $X$, $Y$ are the horizontal and vertical coordinate sets of all points in the Gaussian distribution, respectively, $P_{ix}$ and $P_{iy}$ are the horizontal and vertical coordinates of $J_i$ in the image, and $\sigma_i$ is the standard deviation.

**Parts**. We generate an elliptical Gaussian for each bone to represent its appearance features and add them to get the skeleton's appearance representation. The two joints of the i-th bone $S_i$ are $J_{i1}$ and $J_{i2}$ respectively, and the direction of the bone Gaussian heatmap is consistent with the direction of $J_{i1}$ and $J_{i2}$. The Gaussian heatmap of the bone can be formulated as:

$$H_{S_i} = e^{(-\frac{(X - C_{ix})^2}{2\sigma_{ix}^2} + \frac{(Y - C_{iy})^2}{2\sigma_{iy}^2})}$$ (8)

where $X$, $Y$ are the horizontal and vertical coordinate sets of all points in the Gaussian distribution, respectively, $C_{ix}$ and $C_{iy}$ are the horizontal and vertical coordinates of the center point $C_i$ of the i-th bone $S_i$, and $\sigma_{ix}$ is proportional to the Euclidean distance $||J_{i1} - J_{i2}||$ and set $\sigma_{iy} = k\sigma_{ix}$, where $k = 0.4$.

**Object**. For single-person pose estimation, we need a box to locate the position of a person and it usually comes from the result of object detection or manual annotation. The center

of the box is regarded as the center $C_b$ of the Gaussian distribution and then the Gaussian heatmap of the entire body is expressed as:

$$H_B = e^{(-\frac{(X-C_{bx})^2}{2\sigma_{bx}^2} + \frac{(Y-C_{by})^2}{2\sigma_{by}^2})} \tag{9}$$

where $X$, $Y$ are the horizontal and vertical coordinate sets of all points in the Gaussian distribution, respectively, $C_{bx}$ and $C_{by}$ are the horizontal and vertical coordinates of $C_b$, and $\sigma_{bx}$ and $\sigma_{by}$ are proportional to the maximum difference in the horizontal and vertical coordinates of annotated joints, respectively.

### 3.4 Mutilscale context and prior fusion modules

**Mutilscale context module**. This module is mainly used to model the context relations and get the context information of each structure. Intuitively, the parts of our body are interconnected, just different in strength and weakness. Therefore, it is very necessary to model the relation between various parts.

Considering that the scale of each person in the image is variable, we alleviate this problem by computing context information at different scales. As shown in Fig. 4(a), we first down-sample the feature maps $F_J, F'_{Ji} \in R^{C \times H \times W}$, where $i \in 1, 2..., N$, and get feature maps of different scales. Then the context information of different scales can be obtained through the calculation of the context module. Finally, the context information of different scales is up-sampled to the same resolution $64 \times 64$ and added to obtain multiscale context information. The context module mainly calculates cosine similarity of each element in two inputs. They are first reshaped into $a, a' \in R^{L \times C}, L = (H/n) \times (W/n)$. The context relations can be expressed as:

$$s = a \times (a')^T \tag{10}$$

where $\times$ represents matrix multiplication and $s \in R^{L \times L}$. After normalization, the context information $a_s$ is obtained,

$$a_s = a^T \times s \tag{11}$$

where $a_s \in R^{C \times L}$. It can be seen that the context information comes from $F_J$ or $F_S$ in the bottom-up stage after filtering and combining elements. The context relations and information of each skeleton part are available in the same way.

**Prior fusion module**. As shown in Fig. 4(b), this module has two inputs, such as $F_{J1}$ and coarse prediction results belonging to part 1 of joints. The coarse prediction results from the bottom-up stage are fused with the feature map after the first RB module in the corresponding branches of the top-down stage after raising the dimension to preserve the coarse prediction results.

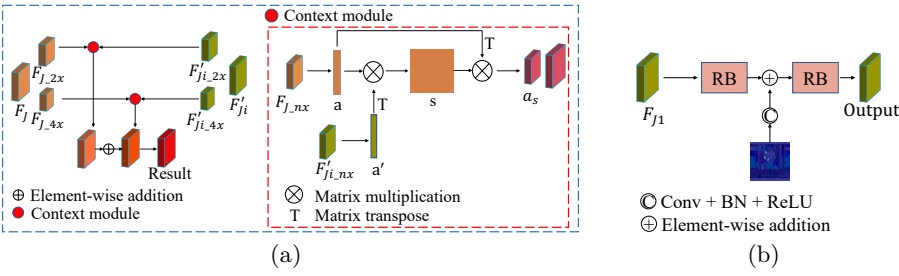

(a)                    (b)

Figure 4: (a) The design of the multiscalse context module. Taking joints as an example, $F_J$ is a feature map of all joints and $F'_{Ji}$ is a feature map of the i-part joint. The core part of the multiscale context module is the context module in the red dotted box. It is responsible for obtaining context relations and information by computing the cosine similarity of each element in the $F_{J\_nx}$ and $F'_{Ji\_nx}$, where $n$ is the downsampling multiple. (b) The design of the prior fusion module. RB is a residual block (He et al., 2016).

Table 1: Comparisons on the COCO validation set.

| Method | Backbone | Input size | MAP | MAR |
|---|---|---|---|---|
| SimpleBaseline (Xiao et al., 2018) | ResNet-152 | 256×192 | 72.0 | 77.8 |
| FEMSFF (Cao et al., 2023) | ResNet-101 | 256×192 | 72.5 | - |
| MSPENet (Xu et al., 2023) | ResNet-50 | 256×192 | 72.7 | 78.3 |
| HPnet (Li et al., 2023) | ResNet-152 | 256×192 | 73.7 | - |
| GMSFF&SMICM (Zhao et al., 2021) | HRNet-W32 | 256×192 | 74.9 | 80.3 |
| HR-ARNet (Wang et al., 2021) | HRNet-W32 | 256×192 | 74.9 | 80.3 |
| EMpose (Yue et al., 2021) | HRNet-W32 | 256×192 | 75.0 | 80.2 |
| OASNet (Zhou et al., 2020) | HRNet-W32 | 256×192 | 75.0 | 80.4 |
| DLIFP (Zhang & Chen, 2023) | HRNet-W32 | 256×192 | 75.0 | - |
| HRNet (Sun et al., 2019) | HRNet-W32 | 256×192 | 74.4 | 79.8 |
| VITPose-B (Xu et al., 2022) | HRNet-W32 | 256×192 | 75.8 | 81.1 |
| Ours | HRNet-W32 | 256×192 | 75.1 | 80.2 |
| FEM&MSFF (Cao et al., 2023) | ResNet-101 | 384×288 | 74.5 | - |
| SimpleBaseline (Xiao et al., 2018) | ResNet-152 | 384×288 | 74.3 | 79.7 |
| HPnet (Li et al., 2023) | ResNet-152 | 384×288 | 75.6 | - |
| HR-ARNet (Wang et al., 2021) | HRNet-W32 | 384×288 | 75.9 | 90.9 |
| HRNet (Sun et al., 2019) | HRNet-W32 | 384×288 | 75.8 | 81.0 |
| Ours | HRNet-W32 | 384×288 | **76.3** | 81.0 |

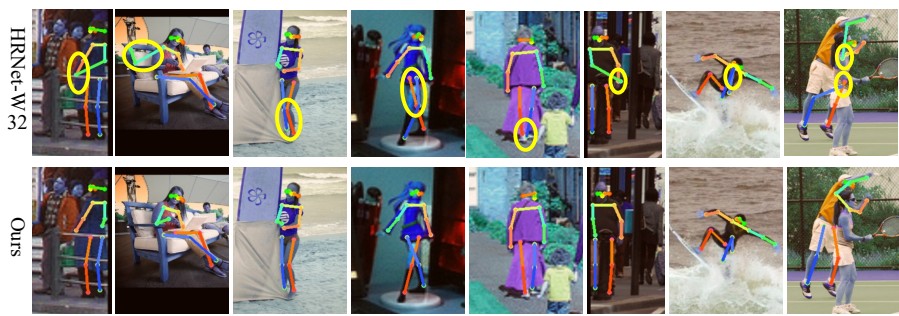

Figure 5: The first line is the result of HRNet and the second line is the result of our method. Obvious erroneous predictions are marked with a yellow oval.

## 4 Experiments

### 4.1 Datasets and evaluation methods

**Datasets**. Our approach is trained and tested on two HPE benchmark datasets: the COCO keypoint detection, Crowdpose and MPII Human Pose datasets. There are over 200k images and 250k person instances labeled with 17 joints in the COCO keypoint detection dataset, with 57k images of training, 5k images of validation, and 20k images of testing. There are about 25k images and 40k annotation samples with 16 joints per instance in the MPII Human Pose dataset, with 28k of training and 11k of testing.

**Evaluation methods**. For COCO or Crowdpose, we use the mean average precision (MAP) and mean average recall (MAR) as evaluation indicators. For MPII, the PCKh score is used as evaluation indicators.

### 4.2 Implementation details

We follow the top-down method for HPE. The training samples are the cropped images with single person. For the COCO keypoint detection dataset, all input images are resized into $256 \times 192$ or $384 \times 288$ resolution. In verifying and testing, we use the detected person

Table 2: Comparisons on the COCO test-dev set.

| Method | Backbone | Input size | MAP | MAR |
|---|---|---|---|---|
| FEM&MSFF (Cao et al., 2023) | ResNet-50 | 256×192 | 71.6 | - |
| MSPENet (Xu et al., 2023) | ResNet-50 | 256×192 | 72.2 | 77.8 |
| DLIFP (Zhang & Chen, 2023) | HRNet-W32 | 256×192 | 73.8 | - |
| TokenPose-Base (Li et al., 2021) | HRNet-W32 | 256×192 | 74.0 | 79.1 |
| EMpose (Yue et al., 2021) | HRNet-W32 | 256×192 | 73.8 | 79.1 |
| HR-ARNet (Wang et al., 2021) | HRNet-W32 | 256×192 | 73.9 | 79.3 |
| OKS-net (Zhao et al., 2020) | HRNet-W32 | 256×192 | 73.9 | 79.3 |
| HRNet (Sun et al., 2019) | HRNet-W32 | 256×192 | 73.5 | 78.9 |
| ViTPose-B (Xu et al., 2022) | ViT-B | 256×192 | 75.1 | 78.3 |
| Ours | HRNet-W32 | 256×192 | 74.2 | 79.3 |
| CPN (Chen et al., 2018) | ResNet | 384×288 | 72.1 | 78.5 |
| CPN (ensemble) (Chen et al., 2018) | ResNet | 384×288 | 73.0 | 79.0 |
| FEM&MSFF (Cao et al., 2023) | ResNet-50 | 384×288 | 73.3 | - |
| SimpleBaseline (Xiao et al., 2018) | ResNet-152 | 384×288 | 73.7 | 79.0 |
| OKS-net (Zhao et al., 2020) | HRNet-W32 | 384×288 | 75.2 | 80.4 |
| GLCFBNet (Zou et al., 2023) | 8-stage-hg | 384×288 | 75.3 | 80.7 |
| HRNet (Sun et al., 2019) | HRNet-W32 | 384×288 | 74.9 | 80.1 |
| Ours | HRNet-W32 | 384×288 | **75.4** | 80.3 |

Table 3: Comparisons on CrowdPose test set with YOLOv3 (Redmon & Farhadi, 2018) human detector. * denotes using a stronger Faster RCNN (Chen & Gupta, 2017) detector.

| Method | Backbone | Input size | MAP |
|---|---|---|---|
| MIPNet (Khirodkar et al., 2021) | ResNet-101 | 384×288 | 68.1 |
| SimpleBaseline (Xiao et al., 2018) | ResNet-152 | 256×192 | 65.6 |
| HRNet (Sun et al., 2019) | HRNet-W32 | 256×192 | 67.5 |
| HRNet* (Sun et al., 2019) | HRNet-W48 | 384×288 | 69.3 |
| Ours | HRNet-W32 | 256×192 | 68.5 |
| Ours | HRNet-W32 | 384×288 | **70.4** |

boxes (Xiao et al., 2018). For the MPII dataset, all input images are resized into $256 \times 256$ resolution. In verifying and testing, we use the provided person boxes and a six-scale pyramid testing method is used (Yang et al., 2017b). Other training and testing strategies are consistent with HRNet. All experiments are finished on two 24GB NVIDIA GeForce RTX 3090 GPUs.

### 4.3 Benchmark results

**Results on COCO keypoint detection task**. The results of our method and other state-of-the-art methods on the validation and test-dev sets are shown in table 1 and 2, respectively. Our network, trained with the input size $256 \times 192$ and $384 \times 288$, achieves 75.1 and 76.3 MAP scores on the validation set and 74.2 and 75.4 MAP scores on the test-dev set, both of which are 0.7 and 0.5 higher than the baseline HRNet respectively, outperforming other methods with the same input size. The visualization results are shown in Fig. 5, which proves that our model can perform better in complex situations. The comparison of network complexity as shown in table 4, since there are multiple branches in our network, the number of parameters increases, but the results are improved to a certain extent.

**Results on Crowdpose benchmark and MPII benchmark**. Table 3 shows that our network achieves 68.5 MAP and 70.4 MAP on CrowdPose test set, which higher than the baseline HRNet. At the same time, our result is better than other methods. Table 5 shows the PCKh@0.5 results of our method and other state-of-the-art methods on the MPII test set. Our network achieves a 92.0 PKCh@0.5 score and outperforms other methods.

Table 4: Complexity comparison and the MAP is obtained on the COCO test-dev set.

| Method | Backbone | Param | MAP |
|---|---|---|---|
| HRNet (Sun et al., 2019) | HRNet-W32 | 28.5M | 74.9 |
| ViTPose-B (Xu et al., 2022) | ViT-B | 86M | 75.1 |
| Ours | HRNet-W32 | 81.8M | **75.4** |

Table 5: Comparisons of PCKh@0.5 scores on the MPII test set.

| Method | Head | Sho. | Elb. | Wri. | Hip | Knee | Ank. | Mean |
|---|---|---|---|---|---|---|---|---|
| De Bem et al. (2018) | 97.7 | 95.0 | 88.1 | 83.4 | 97.9 | 82.1 | 78.7 | 88.1 |
| Newell et al. (2016) | 98.2 | 96.3 | 91.2 | 87.1 | 90.1 | 87.4 | 83.6 | 90.9 |
| Luvizon et al. (2019) | 98.1 | 96.6 | 92.0 | 87.5 | 90.6 | 88.0 | 82.7 | 91.2 |
| Yue et al. (2021) | 98.3 | 96.6 | 91.9 | 87.8 | 90.5 | 88.2 | 84.4 | 91.4 |
| Wang et al. (2021) | 98.3 | 96.7 | 92.4 | 88.5 | 90.4 | 88.3 | 84.4 | 91.6 |
| Zou et al. (2023) | 98.4 | 96.7 | 92.2 | 88.0 | 91.2 | 88.9 | 84.8 | 91.8 |
| Chou et al. (2018) | 98.2 | 96.8 | 92.2 | 88.0 | 91.3 | 89.1 | 84.9 | 91.8 |
| Ryou et al. (2019) | 98.6 | 96.6 | 92.3 | 87.8 | 90.8 | 88.8 | 86.0 | 91.9 |
| Chen et al. (2019) | 98.1 | 96.5 | 92.5 | 88.5 | 90.2 | 89.6 | 86.0 | 91.9 |
| HRNet-W32 | 97.9 | 96.5 | 92.3 | 88.2 | 90.9 | 88.1 | 83.8 | 91.5 |
| Ours | 98.4 | 96.7 | 92.8 | 88.6 | 91.1 | 89.3 | 84.1 | **92.0** |

Table 6: Ablation study of all modules in our network. Comparisons of PCKh@0.5 scores on the MPII test set.

| Method | Head | Sho. | Elb. | Wri. | Hip | Knee | Ank. | Mean |
|---|---|---|---|---|---|---|---|---|
| w/o context | 96.5 | 96.6 | 92.7 | 88.4 | 91.0 | 89.2 | 85.1 | 91.7 |
| w/o prior | 98.3 | 96.5 | 92.5 | 88.3 | 90.8 | 89.0 | 84.4 | 91.8 |
| w/o prior and context | 97.0 | 96.7 | 92.5 | 88.6 | 90.7 | 89.0 | 83.5 | 91.5 |
| w/o mutilscale | 98.4 | 96.6 | 92.6 | 88.4 | 90.7 | 89.5 | 84.8 | 91.9 |
| Ours | 98.4 | 96.7 | 92.8 | 88.6 | 91.1 | 89.3 | 84.1 | **92.0** |

4.4   Ablation study and analysis

**Modules performance**. To verify the validity of each module, our ablation experiments are performed on the MPII test set. Mean PCKh@0.5 is used as the evaluation metric. All results are obtained with the input size of $256 \times 256$. Table 6 shows the PCKh@0.5 results of our ablative experiment. Without context information, the PKCh@0.5 score is reduced by 0.3, without prior information, the PKCh@0.5 score is reduced by 0.2 and without the above two modules, the PKCh@0.5 score is reduced by 0.5. It can be seen that these modules are valid, especially the context module.

**Multiscale performance in context**. Since the scales of the people in the image are different, the context information should also be multiscale so that the model can be more robust to the scale transformation of people. Table 6 also shows the PKCh@0.5 score is reduced by 0.1 without multiscale, and this proves that multiscale is effective.

5   Conclusion

Inspired by parse graphs, we design a hierarchical network according to the parse graph of body structure. Experiments show that the hierarchical structures and context information provide structural and relational between structures constraints, which makes the network solution space smaller and the prediction results more reasonable and accurate. We hope that in future work, this kind of method can become more mature and be used in more visual tasks.

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
