# OpenReview forum: "Human Pose Estimation via Parse Graph of Body Structure"
_ICLR.cc/2024/Conference — ICLR 2024 Conference Withdrawn Submission_

### Official Review · Reviewer_6Yh6 · 2023-10-16

**Soundness:** 2 fair
**Presentation:** 3 good
**Contribution:** 1 poor
**Rating:** 3
**Confidence:** 5

**Summary:**

This paper aims to improve the human pose estimation task and proposes a method to model the context relations and hierarchical structure in the parsing graph of body structure by using convolutional neural networks.

**Strengths:**

1. The writing is good and easy to follow.
2. The paper model the context relations and hierarchical structure in the body structure.

**Weaknesses:**

1. The experimental results of this paper are rather poor, and it does not cite some of the recent methods: DARK[1], UDP[2], ViTPose[3], PCT[4].
2. The authors claim in the Introduction that it should be the first time that the context relations and hierarchical structure in the parse graph of body structure are explicitly modeled by convolutional neural networks (CNNs) for HPE. However, many papers from 2018 have already made such attempts: [5,6,7] and the authors did not cite any of them.
3. This paper does not conduct any ablation study.

[1]. Feng Zhang, Xiatian Zhu, Hanbin Dai, Mao Ye, and Ce Zhu. Distribution-aware coordinate representation for human pose estimation. In CVPR 2020.

[2]. Junjie Huang, Zheng Zhu, Feng Guo, and Guan Huang. The devil is in the details: Delving into unbiased data processing for human pose estimation. In CVPR 2020.

[3]. Yufei Xu, Jing Zhang, Qiming Zhang, and Dacheng Tao. Vitpose: Simple vision transformer baselines for human pose estimation.

[4]. Zigang Geng, Chunyu Wang, Yixuan Wei, Ze Liu, Houqiang Li, Han Hu. Human Pose as Compositional Tokens. In CVPR 2023.

[5]. Wei Yang, Wanli Ouyang, Hongsheng Li, and Xiaogang Wang. End-to-end learning of deformable mixture of parts and deep convolutional neural networks for human pose estimation. In CVPR 2016.

[6]. Hong Zhang, Hao Ouyang, Shu Liu, Xiaojuan Qi, Xiaoyong Shen, Ruigang Yang, and Jiaya Jia. Human pose estimation with spatial contextual information. In CoRR 2019.

[7]. Xiao Chu, Wanli Ouyang, Hongsheng Li, and Xiaogang Wang. Structured feature learning for pose estimation. In CVPR 2016.

**Questions:**

I think bottom-up and top-down are two specific terms in the context of human pose estimation and using them in the description of your method may not be quite appropriate.

---

> ### Author Response · Authors · 2023-11-21
> **Experiments and Innovative Descriptions**
>
> Thank you for taking the time to read my paper and provide me with helpful comments. Below is my reply.
> ## Innovative Descriptions
> We carefully review related work and revise the contributions of our paper. The specific modifications are as follows:
> 1. We design a novel and valid architecture to learn the hierarchical structure and context relations in the parse graph via convolutional neural networks (CNNs)
> ## Related Work
> Human pose estimation using parse graphs is novel and interpretable. Although similar ideas have been used in [a-b], Tang et al. [c] argue that they have difficulty describing the complex compositional relationships among body parts due to some unreasonable assumptions. So [c] models the hierarchical structure by supervision of heatmaps in a bottom-up\top-down way. However, [c-f] always ignore one of the hierarchical structure and the context relations or learn them implicitly. To sum up, we model the hierarchical structure and context relations in the parse graph by supervision of heatmaps in a bottom-up\top-down way. So our work is meaningful and innovative.
> please take a look at the introduction section of our paper for specific modifications.
> ## Experiments Descriptions
> We added results and complexity comparisons with recent methods, such as VitPose [g]. Moreover, we add the experiments on the CrowdPose [h] dataset to verify the validity of our method. Brief results are as follows:
> ### Results on CrowdPose test set
> | Method    | Backbone     | Input size    | MAP|
> | -------- | -------- | -------- |--------|
> | HRNet [i] | HRNet-w32 | 256x192 |67.5|
> | HRNet* | HRNet-w48 | 384x288 |69.3|
> | Ours | HRNet-w32 | 256x192 |68.5|
> | Ours | HRNet-w32 | 384x288 |70.4|
>
> ### The Complexity comparison with other methods and MAP is obtained on the COCO test-dev set.
> | Method    | Backbone     | Parma    | MAP|
> | -------- | -------- | -------- |--------|
> | HRNet | HRNet-w32 | 28.5M |74.9|
> | ViTPose | Vit-B | 86M |75.1|
> | Ours | HRNet-w32 | 81.8M |75.4|
>
> please take a look at Tables 3 and 4 in our paper for specific modifications.
> ## About Ablation Study
> we included the results of the ablation study in our initial paper submission and the results of the ablation study in Table 6 of our paper.
> ## Future
> There are areas worth improving in our framework. First, we can introduce better backbones, such as VitPose [g], and then we can introduce [j-k]. We think this may greatly improve our results, and we will conduct related experiments later.
>
> [a] Park S, Nie B X, Zhu S C. Attribute and-or grammar for joint parsing of human pose, parts and attributes[J]. IEEE transactions on pattern analysis and machine intelligence, 2017, 40(7): 1555-1569.
>
> [b] Wang, Y., Tran, D., Liao, Z.: Learning hierarchical poselets for human parsing. In: IEEE Conference on Computer Vision and Pattern Recognition. (2011) 1705–1712
>
> [c] Tang W, Yu P, Wu Y. Deeply learned compositional models for human pose estimation[C]//Proceedings of the European conference on computer vision (ECCV). 2018: 190-206.
>
> [d] Chen X, Mottaghi R, Liu X, et al. Detect what you can: Detecting and representing objects using holistic models and body parts[C]//Proceedings of the IEEE conference on computer vision and pattern recognition. 2014: 1971-1978.
>
> [e] Chen X, Yuille A L. Articulated pose estimation by a graphical model with image dependent pairwise relations[J]. Advances in neural information processing systems, 2014, 27.
>
> [f] Chu X, Ouyang W, Wang X. Crf-cnn: Modeling structured information in human pose estimation[J]. Advances in neural information processing systems, 2016, 29.
>
> [g] Xu Y, Zhang J, Zhang Q, et al. Vitpose: Simple vision transformer baselines for human pose estimation[J]. Advances in Neural Information Processing Systems, 2022, 35: 38571-38584.
>
> [h] Li J, Wang C, Zhu H, et al. Crowdpose: Efficient crowded scenes pose estimation and a new benchmark[C]//Proceedings of the IEEE/CVF conference on computer vision and pattern recognition. 2019: 10863-10872.
>
> [i] Sun K, Xiao B, Liu D, et al. Deep high-resolution representation learning for human pose estimation[C]//Proceedings of the IEEE/CVF conference on computer vision and pattern recognition. 2019: 5693-5703.
>
> [j] Feng Zhang, Xiatian Zhu, Hanbin Dai, Mao Ye, and Ce Zhu. Distribution-aware coordinate representation for human pose estimation. In CVPR 2020.
>
> [k] Junjie Huang, Zheng Zhu, Feng Guo, and Guan Huang. The devil is in the details: Delving into unbiased data processing for human pose estimation. In CVPR 2020.

---

### Official Review · Reviewer_qq5T · 2023-10-25

**Soundness:** 3 good
**Presentation:** 2 fair
**Contribution:** 2 fair
**Rating:** 3
**Confidence:** 4

**Summary:**

This paper proposes a method to capture both the visual features and body structure for human pose estimation. It first predicts heatmaps of primitives, parts and body at the bottom-up stage, then refine the outputs of different levels at the top-down stage. The estimation results are improved after refinement.

**Strengths:**

The structure of human body is explicitly exploited in the manner of bottom-up and top-down stages for the task of human pose estimation. The experiments on the COCO and MPII datasets show the superiority of the proposed method.

**Weaknesses:**

The contribution of this paper is somewhat incremental. The structure of human body has been widely used as a prior for the task of human pose estimation. The detailed comparison including theoretical analysis and experimental results should be provided to validate the superiority of the proposed method over the previous approaches (e.g., how to avoid the over-fitting problem as mentioned in the introduction section).

The applied parse graph is critical to this work. The effect of different parse graphs and the effectiveness of the parse graph applied in this work should be evaluated in the experiments, which is missing in the current version.

The computational complexity of the proposed approach is expected to be compared with those of the existing methods. The proposed method can be regarded as a refinement module added to HRNet, so the model complexity is a concern.

**Questions:**

See weaknesses.

**Details Of Ethics Concerns:**

The template used in this paper seems to be a little different from the official ICLR template.

---

> ### Author Response · Authors · 2023-11-21
> **The Introduction of Contribution , Complexity Comparison and the Explanation of Method Superiority**
>
> Thank you for taking the time to read my paper and provide me with helpful comments. Below is my reply.
> ## Contribution
> We carefully review related work and revise the contributions of our paper. The specific modifications are as follows:
> 1. We design a novel and valid architecture to learn the hierarchical structure and context relations in the parse graph via convolutional neural networks (CNNs)
> ## Related Work
> Human pose estimation using parse graphs [a] is novel and interpretable. Although similar ideas have been used in [b-e], Tang et al. [f] argue that they have difficulty describing the complex compositional relationships among body parts due to some unreasonable assumptions. So [f] models the hierarchical structure by supervision of heatmaps in a bottom-up\top-down way. However, [f-j] always ignore one of the hierarchical structure and the context relations or learn them implicitly. To sum up, we model the hierarchical structure and context relations in the parse graph by supervision of heatmaps in a bottom-up\top-down way. So our work is meaningful and innovative.
>
> ## Experiments
> Moreover, we add the experiments on CrowdPose [k] datasets and the experiments of complexity comparison to verify the validity of our method. Brief results are as follows:
> ### Results on CrowdPose test set
> | Method    | Backbone     | Input size    | MAP|
> | -------- | -------- | -------- |--------|
> | HRNet [l] | HRNet-w32 | 256x192 |67.5|
> | HRNet* | HRNet-w48 | 384x288 |69.3|
> | Ours | HRNet-w32 | 256x192 |68.5|
> | Ours | HRNet-w32 | 384x288 |70.4|
>
> ### The Complexity comparison with other methods and MAP is obtained on the COCO test-dev set.
> | Method    | Backbone     | Parma    | MAP|
> | -------- | -------- | -------- |--------|
> | HRNet | HRNet-w32 | 28.5M |74.9|
> | ViTPose [m]| Vit-B | 86M |75.1|
> | Ours | HRNet-w32 | 81.8M |75.4|
>
> please take a look at the introduction section, and Tables 3 and 4 for specific modifications.
> ## Key Issue of Different Parse Graphs
> We noticed that you mentioned a key issue regarding the comparison of different parse graphs. We believe that different human bodies correspond to different parse graphs. Obtaining the correct parse graph helps complete the task of human pose estimation. This study will be carried out in our subsequent research work. However, in this paper, we just want to demonstrate the effectiveness of parse graphs. Therefore, a more general parse graph of human body structure is made (see Fig. 1(a) in this paper), and human pose estimation is performed based on it.
>
> [a] Zhu S C, Mumford D. A stochastic grammar of images[J]. Foundations and Trends® in Computer Graphics and Vision, 2007, 2(4): 259-362.
>
> [b] Park S, Nie B X, Zhu S C. Attribute and-or grammar for joint parsing of human pose, parts and attributes[J]. IEEE transactions on pattern analysis and machine intelligence, 2017, 40(7): 1555-1569.
>
> [c] Wang, Y., Tran, D., Liao, Z.: Learning hierarchical poselets for human parsing. In: IEEE Conference on Computer Vision and Pattern Recognition. (2011) 1705–1712
>
> [d] Rothrock, B., Park, S., Zhu, S.C.: Integrating grammar and segmentation for human pose estimation. In: IEEE Conference on Computer Vision and Pattern Recognition. (2013) 3214–3221
>
> [e] Johnson S, Everingham M. Clustered pose and nonlinear appearance models for human pose estimation[C]//bmvc. 2010, 2(4): 5.
>
> [f] Tang W, Yu P, Wu Y. Deeply learned compositional models for human pose estimation[C]//Proceedings of the European conference on computer vision (ECCV). 2018: 190-206.
>
> [g] Chen X, Mottaghi R, Liu X, et al. Detect what you can: Detecting and representing objects using holistic models and body parts[C]//Proceedings of the IEEE conference on computer vision and pattern recognition. 2014: 1971-1978.
>
> [h] Chen X, Yuille A L. Articulated pose estimation by a graphical model with image dependent pairwise relations[J]. Advances in neural information processing systems, 2014, 27.
>
> [i] Chu X, Ouyang W, Wang X. Crf-cnn: Modeling structured information in human pose estimation[J]. Advances in neural information processing systems, 2016, 29.
>
> [j] Yang Y, Ramanan D. Articulated human detection with flexible mixtures of parts[J]. IEEE transactions on pattern analysis and machine intelligence, 2012, 35(12): 2878-2890.
>
> [k] Li J, Wang C, Zhu H, et al. Crowdpose: Efficient crowded scenes pose estimation and a new benchmark[C]//Proceedings of the IEEE/CVF conference on computer vision and pattern recognition. 2019: 10863-10872.
>
> [l] Sun K, Xiao B, Liu D, et al. Deep high-resolution representation learning for human pose estimation[C]//Proceedings of the IEEE/CVF conference on computer vision and pattern recognition. 2019: 5693-5703.
>
> [m] Xu Y, Zhang J, Zhang Q, et al. Vitpose: Simple vision transformer baselines for human pose estimation[J]. Advances in Neural Information Processing Systems, 2022, 35: 38571-38584.

---

> > ### Comment · Reviewer_qq5T · 2023-12-01
> >
> > The reviewer appreciates the authors' response. I have read all the comments from reviewers and rebuttal from authors. However, due to the concerns about technical novelty and experimental results, I believe this paper has not been ready for publication at this stage.

---

### Official Review · Reviewer_8zRF · 2023-10-26

**Soundness:** 3 good
**Presentation:** 2 fair
**Contribution:** 2 fair
**Rating:** 5
**Confidence:** 3

**Summary:**

This paper proposes an new approach for 2D pose estimation with a parse graph. The proposed method, as the author states, is the first time that uses CNN to model the contextual relations and hierarchical structure while simultaneously employing both top-down and bottom-up approaches in the network.This work enhances the reasonableness of predictions and achieves superior results on the COCO keypoint detection and MPII human pose datasets.

**Strengths:**

1. The idea of representing the human body's structure using parse graphs is simple and intuitive.
2. The paper is well-written with good illustrations and visualizations of results for important cases

**Weaknesses:**

1. The innovation in this paper may not be entirely clear. It seems to me that proposed method is a combination of ideas from previous works. The paper could benefit from further clarifying the differences and new contributions compared to existing methods.
2. The performance of the proposed method is not impressive and the experiments do not compare with SOTA methods, e.g., VitPose [1].

[1] Xu Y, Zhang J, Zhang Q, et al. Vitpose: Simple vision transformer baselines for human pose estimation[J]. Advances in Neural Information Processing Systems, 2022, 35: 38571-38584.

**Questions:**

See Weaknesses

---

> ### Author Response · Authors · 2023-11-20
> **The Details of Our Innovation and New Experimental Results**
>
> Thank you for taking the time to read my paper and provide me with helpful comments. Below is my reply.
> ## Our Innovation
> we carefully review related work and revise the contributions of our paper. The specific modifications are as follows:
> 1. We design a novel and valid architecture to learn the hierarchical structure and context relations in the parse graph via convolutional neural networks (CNNs)
>
> Please refer to the introduction section for details in the latest submitted paper.
> ## Experiments
> Moreover, we add the experiments on CrowdPose [b] datasets and the experiments of complexity comparison to verify the validity of our method. Brief results are as follows:
> ### Results on CrowdPose test set
> | Method    | Backbone     | Input size    | MAP|
> | -------- | -------- | -------- |--------|
> | HRNet [c] | HRNet-w32 | 256x192 |67.5|
> | HRNet* | HRNet-w48 | 384x288 |69.3|
> | Ours | HRNet-w32 | 256x192 |68.5|
> | Ours | HRNet-w32 | 384x288 |70.4|
>
> ### The Complexity comparison with other methods and MAP is obtained on the COCO test-dev set.
> | Method    | Backbone     | Parma    | MAP|
> | -------- | -------- | -------- |--------|
> | HRNet | HRNet-w32 | 28.5M |74.9|
> | ViTPose [a]| Vit-B | 86M |75.1|
> | Ours | HRNet-w32 | 81.8M |75.4|
>
> please take a look at the introduction section, and Tables 3 and 4 for specific modifications.
>
> [a] Xu Y, Zhang J, Zhang Q, et al. Vitpose: Simple vision transformer baselines for human pose estimation[J]. Advances in Neural Information Processing Systems, 2022, 35: 38571-38584.
>
> [b] Li J, Wang C, Zhu H, et al. Crowdpose: Efficient crowded scenes pose estimation and a new benchmark[C]//Proceedings of the IEEE/CVF conference on computer vision and pattern recognition. 2019: 10863-10872.
>
> [c] Sun K, Xiao B, Liu D, et al. Deep high-resolution representation learning for human pose estimation[C]//Proceedings of the IEEE/CVF conference on computer vision and pattern recognition. 2019: 5693-5703.

---

### Official Review · Reviewer_y1Ri · 2023-10-28

**Soundness:** 2 fair
**Presentation:** 2 fair
**Contribution:** 2 fair
**Rating:** 3
**Confidence:** 5

**Summary:**

The paper proposes a hierarchical network for human pose estimation based on the parse graph of body structure. The network consists of a top-down stage and a bottom-up stage, which work together to refine the pose estimation results. The top-down stage utilizes context information and prior fusion to improve the accuracy of the predictions, while the bottom-up stage generates coarse predictions based on the input image. Experiments are conducted on COCO and MPII. Ablation studies show the effectiveness of the proposed module.

**Strengths:**

1. Solving human pose estimation with graph-based representation is reasonable and important.
2. The paper provided the quantitative evaluation, ablation study, and qualitative evaluation.

**Weaknesses:**

**Weakness:**

1.	Overclaim. “As far as we know, this should be the first time that the context relations and hierarchical structure in the parse graph of body structure are explicitly modeled by convolutional neural networks (CNNs) for HPE.” This statement is overclaimed. In fact, there are a number of papers exploiting hierarchical structure of the human body. For example, [a] proposes hierarchical graph grouping for a more challenging multi-person pose estimation problem. [b-h] are works that apply graphical model for single-person human pose modeling. Please discuss the relationship of these works and compare with them.
2.	The novelty of this paper is concerning, especially considering the missing discussion of the related works [a-h].
3.	Somewhat insufficient experimental validation.
a)	The experiments are conducted on COCO and MPII. The accuracy of MPII is near saturated (over 90 PCKh). Please consider using other challenging datasets, e.g. CrowdPose[i].
b)	State-of-the-art methods are not compared. It does not achieve the state-of-the-art performance. For example, ViTPose[j]. And most importantly, PGNN[i] is also a graph-based model which achieves 92.5 on MPII test, better than this submission (92.0) but not reported.
c)	Computational efficiency: The paper may not address the computational complexity or efficiency of the proposed method, which could be a concern in practical applications. Please report the FLOPs and runtime speed, and compare it with other approaches (e.g. HRNet).
d)     The ablation study is conducted on MPII, where the performance gap is insignificant. Please consider using COCO for ablation study.

**Minor:**

1.	The paper requires careful proof reading.

2.	The format of the submission does not meet the standards of ICLR2024. Please use the template provided.

3.	There are repeated entries in the References. Please correct them.

4.	Table1 & Table2, it is suggested noting “HRNet-W32” instead of “W-32” for the backbone for easier understanding.


[a] Jin S, Liu W, Xie E, et al. Differentiable hierarchical graph grouping for multi-person pose estimation[C]//Computer Vision–ECCV 2020: 16th European Conference, Glasgow, UK, August 23–28, 2020, Proceedings, Part VII 16. Springer International Publishing, 2020: 718-734.

[b] Chen X, Mottaghi R, Liu X, et al. Detect what you can: Detecting and representing objects using holistic models and body parts[C]//Proceedings of the IEEE conference on computer vision and pattern recognition. 2014: 1971-1978.

[c] Chen X, Yuille A L. Articulated pose estimation by a graphical model with image dependent pairwise relations[J]. Advances in neural information processing systems, 2014, 27.

[d] Chu X, Ouyang W, Wang X. Crf-cnn: Modeling structured information in human pose estimation[J]. Advances in neural information processing systems, 2016, 29.

[e] Johnson S, Everingham M. Clustered pose and nonlinear appearance models for human pose estimation[C]//bmvc. 2010, 2(4): 5.

[f] Tompson J J, Jain A, LeCun Y, et al. Joint training of a convolutional network and a graphical model for human pose estimation[J]. Advances in neural information processing systems, 2014, 27.

[g] Yang Y, Ramanan D. Articulated human detection with flexible mixtures of parts[J]. IEEE transactions on pattern analysis and machine intelligence, 2012, 35(12): 2878-2890.

[h] Zhang H, Ouyang H, Liu S, et al. Human pose estimation with spatial contextual information[J]. arXiv preprint arXiv:1901.01760, 2019.

[i] Li J, Wang C, Zhu H, et al. Crowdpose: Efficient crowded scenes pose estimation and a new benchmark[C]//Proceedings of the IEEE/CVF conference on computer vision and pattern recognition. 2019: 10863-10872.

[j] Xu Y, Zhang J, Zhang Q, et al. Vitpose: Simple vision transformer baselines for human pose estimation[J]. Advances in Neural Information Processing Systems, 2022, 35: 38571-38584.

**Questions:**

Please refer to the Weakness section. Please add discussions about other related works and especially highlight the difference.

---

> ### Author Response · Authors · 2023-11-19
> **About the Discussion of Overclaim, Related Works and Experimental Results**
>
> Thank you for taking the time to read my paper and provide me with helpful comments. Below is my reply.
> ## Overclaim
> We carefully analyze the relevant work and modify the contribution points to avoid over-claiming. The specific modifications are as follows:
> 1. We design a novel and valid architecture to learn the hierarchical structure and context relations in the parse graph via convolutional neural networks (CNNs).
>
> Please see the introduction for specific modifications
> ## Related Work
> To be clear, the research content of this paper is single-person pose estimation, so our ideas and methods are different from this kind of method, such as [b]. Although similar ideas have been used in [c-e,j], Tang et al. [f] argue that they have difficulty describing the complex compositional relationships among body parts due to some unreasonable assumptions.  So [f] models the hierarchical structure by supervision of heatmaps in a bottom-up\top-down way. However, [f,g-i,k-l] always ignore one of the hierarchical structures and the context relations or learn them implicitly. To sum up, we model the hierarchical structure and context relations in the parse graph by supervision of heatmaps in a bottom-up\top-down way. So our work is meaningful and innovative.
>
> Please see the introduction for specific modifications.
> ## Experiments
> Moreover, we add the experiments on CrowdPose [a] datasets and the experiments of complexity comparison to verify the validity of our method. Brief results are as follows:
> ### Results on CrowdPose test set
> | Method    | Backbone     | Input size    | MAP|
> | -------- | -------- | -------- |--------|
> | HRNet | HRNet-w32 | 256x192 |67.5|
> | HRNet* | HRNet-w48 | 384x288 |69.3|
> | Ours | HRNet-w32 | 256x192 |68.5|
> | Ours | HRNet-w32 | 384x288 |70.4|
>
> ### The Complexity comparison with other methods and MAP is obtained on the COCO test-dev set.
> | Method    | Backbone     | Parma    | MAP|
> | -------- | -------- | -------- |--------|
> | HRNet | HRNet-w32 | 28.5M |74.9|
> | ViTPose | Vit-B | 86M |75.1|
> | Ours | HRNet-w32 | 81.8M |75.4|
>
> please take a look at the introduction section, and Tables 3 and 4 for specific modifications.
>
> [a] Li J, Wang C, Zhu H, et al. Crowdpose: Efficient crowded scenes pose estimation and a new benchmark[C]//Proceedings of the IEEE/CVF conference on computer vision and pattern recognition. 2019: 10863-10872.
>
> [b] Jin S, Liu W, Xie E, et al. Differentiable hierarchical graph grouping for multi-person pose estimation[C]//Computer Vision–ECCV 2020: 16th European Conference, Glasgow, UK, August 23–28, 2020, Proceedings, Part VII 16. Springer International Publishing, 2020: 718-734.
>
> [c] Park S, Nie B X, Zhu S C. Attribute and-or grammar for joint parsing of human pose, parts and attributes[J]. IEEE transactions on pattern analysis and machine intelligence, 2017, 40(7): 1555-1569.
>
> [d] Wang, Y., Tran, D., Liao, Z.: Learning hierarchical poselets for human parsing. In: IEEE Conference on Computer Vision and Pattern Recognition. (2011) 1705–1712
>
> [e] Rothrock, B., Park, S., Zhu, S.C.: Integrating grammar and segmentation for human pose estimation. In: IEEE Conference on Computer Vision and Pattern Recognition. (2013) 3214–3221
>
> [f] Tang W, Yu P, Wu Y. Deeply learned compositional models for human pose estimation[C]//Proceedings of the European conference on computer vision (ECCV). 2018: 190-206.
>
> [g] Chen X, Mottaghi R, Liu X, et al. Detect what you can: Detecting and representing objects using holistic models and body parts[C]//Proceedings of the IEEE conference on computer vision and pattern recognition. 2014: 1971-1978.
>
> [h] Chen X, Yuille A L. Articulated pose estimation by a graphical model with image dependent pairwise relations[J]. Advances in neural information processing systems, 2014, 27.
>
> [i] Chu X, Ouyang W, Wang X. Crf-cnn: Modeling structured information in human pose estimation[J]. Advances in neural information processing systems, 2016, 29.
>
> [j] Johnson S, Everingham M. Clustered pose and nonlinear appearance models for human pose estimation[C]//bmvc. 2010, 2(4): 5.
>
> [k] Yang Y, Ramanan D. Articulated human detection with flexible mixtures of parts[J]. IEEE transactions on pattern analysis and machine intelligence, 2012, 35(12): 2878-2890.
>
> [l] Zhang H, Ouyang H, Liu S, et al. Human pose estimation with spatial contextual information[J]. arXiv preprint arXiv:1901.01760, 2019.